# Structural Basis for pH-gating of the K$^+$ channel TWIK1 at the selectivity filter

Toby S. Turney[1,2,3,4], Vivian Li[2,3,4] & Stephen G. Brohawn [2,3,4 ✉]

TWIK1 (K2P1.1, *KCNK1*) is a widely expressed pH-gated two-pore domain K$^+$ channel (K2P) that contributes to cardiac rhythm generation and insulin release from pancreatic beta cells. TWIK1 displays unique properties among K2Ps including low basal activity and inhibition by extracellular protons through incompletely understood mechanisms. Here, we present cryo-EM structures of TWIK1 in lipid nanodiscs at high and low pH that reveal a previously undescribed gating mechanism at the K$^+$ selectivity filter. At high pH, TWIK1 adopts an open conformation. At low pH, protonation of an extracellular histidine results in a cascade of conformational changes that close the channel by sealing the top of the selectivity filter, displacing the helical cap to block extracellular ion access pathways, and opening gaps for lipid block of the intracellular cavity. These data provide a mechanistic understanding for extracellular pH-gating of TWIK1 and illustrate how diverse mechanisms have evolved to gate the selectivity filter of K$^+$ channels.

[1] Biophysics Graduate Program, University of California Berkeley, Berkeley, CA 94720, USA. [2] Department of Molecular & Cell Biology, University of California Berkeley, Berkeley, CA 94720, USA. [3] Helen Wills Neuroscience Institute, University of California Berkeley, Berkeley, CA 94720, USA. [4] California Institute for Quantitative Biosciences (QB3), University of California Berkeley, Berkeley, CA 94720, USA. ✉email: brohawn@berkeley.edu

Two-pore domain K$^+$ channels (K2Ps) generate leak-type currents regulated by diverse chemical and physical stimuli including temperature, membrane tension, signaling lipids, and pH to control the resting membrane potential of cells[1]. K2Ps assemble as homo- or hetero-dimers, with each subunit containing four transmembrane-spanning helices (TM1–TM4), two selectivity filters (SF1–SF2), two pore helices (PH1–PH2), and two extracellular cap helices (CH1–CH2)[1,2]. K2P1.1 or TWIK1 (Tandem of Pore Domains in a Weak Inward Rectifying K$^+$ Channel) was the first K2P to be discovered and is widely expressed, including at high levels in the brain and heart in humans[3].

TWIK1 is a pH-gated K$^+$ channel that responds to extracellular acidification. TWIK1 is predominantly closed at pH 5.5 and open at pH 7.5 with a midpoint pH ≈ 6.7[4]. H122 has been identified as the extracellular proton sensor and is located directly above the selectivity filter. The TWIK1 selectivity filter is unique among K$^+$ channels, having diverged from the canonical T(L/V/I)G(Y/F)G motif to TTGYG for SF1 and TIGLG for SF2[5]. Under conditions of low pH$_{ext}$ and low [K$^+$]$_{ext}$ (<5 mM), TWIK1 reversibly enters a non-selective conductive state rather than close as seen at higher [K$^+$]$_{ext}$[5–7]. This pH$_{ext}$- and [K$^+$]$_{ext}$ -dependent selectivity loss is postulated to underlie the seemingly paradoxical hyperpolarization of kidney and pancreatic β cells upon TWIK1 deletion[5,8] and TWIK1-dependent depolarization of human cardiomyocytes under hypokalemic conditions[6,7].

Currently, structural insight into TWIK1 comes from a crystal structure determined at pH 8[9]. Consistent with the channel being open at high pH, the TWIK1 selectivity filter adopted a conductive conformation with K$^+$ ions bound at canonical sites S0-S4[9]. However, the ion conduction path was blocked in the intracellular cavity by a detergent or lipid acyl chain bound through a lateral membrane opening (or fenestration) between transmembrane helices TM2 and TM4, suggesting the structure represented a closed state[9]. The cavity and lateral membrane opening have since been implicated in regulation of other K2Ps by lipids and small molecules[2]. Acyl chain block of the channel cavity has been observed in structures of TWIK1[9], TRAAK (K2P4.1)[10], and TREK2 (K2P10.1)[11] in detergent micelles and TASK2 (K2P5.1) in lipid nanodiscs[12]. TASK1 (K2P3.1) inhibitors (BAY1000493 and BAY2341237) occupy a similar cavity site to occlude conduction[13], while drug binding in the TREK2 fenestration can either activate (BL-1249)[14] or inhibit (norfluoxetine)[11,15] channel activity through distinct mechanisms. Exactly how the cavity site is involved in TWIK1 gating is unclear[16–18]. Mutational, spectroscopic, and electrophysiological evidence implicate the TWIK1 selectivity filter in gating conformational changes in ways that are distinct from other K$^+$ channels[5,6,19,20].

The structural and mechanistic basis for TWIK1 pH-gating remains unknown. Here, we present cryo-EM structures of TWIK1 in lipid nanodiscs at high and low pH that, together with electrophysiological interrogation of structure-guided mutants, reveal a previously undescribed gating mechanism involving the K$^+$ selectivity filter, extracellular cap, and intracellular cavity of the channel.

## Results

We found that full-length *Rattus norvegicus* TWIK1, which shares 96% sequence identity to human TWIK1, was well-expressed in *Pichia pastoris* and biochemically suitable for structural studies. Rat TWIK1 displayed K$^+$-selective currents with reversal potentials near predicted equilibrium potentials for K$^+$ ($E_{rev}$ = −80.2 ± 2.3 mV and −6.6 ± 0.5 mV in low (2 mM) and high (96 mM) [K$^+$]$_{ext}$, respectively (mean±sem, $n$ = 4 cells)). At high [K$^+$]$_{ext}$, the channel was strongly inhibited (~90%) by low pH$_{ext}$ (Fig. 1a). At low [K$^+$]$_{ext}$,

exposure to low pH$_{ext}$ instead resulted in reduced K$^+$ selectivity indicated by a shift to more positive reversal potentials ($E_{rev}$ = −28.6 ± 3.7 mV (mean ± sem, $n$ = 7)) (Fig. 1b). These properties are consistent with those reported for human TWIK1[3–6]. Similar to previous work[5], all electrophysiological recordings were conducted in a triple mutant background (TWIK1 K274E, I293A, I294A) that eliminates intracellular sequestration, precludes possible SUMOylation, and increases functional expression of the channel[4,21,22].

To capture structures of TWIK1 in open and closed states within a lipid environment, we reconstituted the channel in nanodiscs containing the lipids DOPE, POPC, and POPS (Supplementary Fig. 1) and determined its structure by cryo-EM at pH 7.4 and pH 5.5 to ~3.4 Å resolution (Fig. 1c–e, Supplementary Figs. 2–4 and Table 1). The channel is two-fold symmetric at both pH values and enforcing C$_2$ symmetry of the reconstructions improved map quality. Amino acids W20 to Y281 are clearly defined and modeled in the low pH structure, corresponding to a resolved mass of 59 kDa. The high pH reconstruction is more anisotropic due to preferred orientations of particles (Supplementary Fig. 4a, d). Some regions, particularly the intracellular termini and TM2–TM3 linker, are less well resolved in the high pH map, but amino acids G24 to F280 could still be confidently modeled. TWIK1, like other K2Ps, is a domain-swapped homodimer (Fig. 1d). The initial crystal structure of TWIK1[9] was modeled without a domain swap likely due to poor local resolution of the distal region of the helical cap where the protomers are closely juxtaposed, as was the case for the related K2P TRAAK[23,24].

Consistent with functional data, TWIK1 adopts an open conformation at pH 7.4 (Fig. 2a, c, d) and a closed conformation at pH 5.5 (Fig. 2b–d). The high pH structure shows an unobstructed path from the intracellular to extracellular solutions through the channel cavity, selectivity filter, and bifurcated extracellular tunnels under the helical cap (Figs. 2a, c, d). At low pH, the conduction pathway is closed at three positions. Gating conformational changes seal the top of the selectivity filter and extracellular pathways under the helical cap, forming constrictions with radii of ~0.5 Å and ~0.8 Å that are too small for K$^+$ ions to pass (Fig. 2b–d). The conduction path is additionally occluded at low pH below the intracellular side of the selectivity filter by lipid acyl chains bound within the channel cavity (Fig. 2b, d). The three regions involved in channel closure are considered in turn below.

At high pH, the selectivity filter adopts a nearly four-fold symmetric conformation with K$^+$ ions bound at five positions in sites S0-S4, as observed in other conductive K$^+$ channels and the previously reported TWIK1 crystal structure at pH 8 (Fig. 3a–c, f) (Cα r.m.s.d. within pore helices and selectivity filters = 0.6 Å). The intracellular side of the filter is in a similar conformation at both pH values (Fig. 3a–c), but the extracellular side of the filter is dramatically rearranged at low pH (Fig. 3d). Rotation of Y120 and L228 carbonyls at the top of S1 nearly 180° away from the conduction axis and displacement of the SF1-TM2 and SF2-TM4 linkers disrupts the K$^+$ coordination environment at S1 and S0 sites (Fig. 3e). Due to the change in coordination environment, K$^+$ binding sites S0-S1 are unoccupied at low pH and density corresponding to K$^+$ ions is only observed at positions S2-S4 (Fig. 3g).

Gating conformational changes are further propagated from the top of the selectivity filter (Fig. 3d). At high pH, the residues immediately after the selectivity filter, H122 and D230, project back behind the filter and away from the conduction axis (Fig. 3b, c). Upon protonation, H122 flips upward towards the helical cap (Fig. 3b, c). The adjoining linker region connecting SF1 to TM2 (H122 to D128) kinks outward towards the SF2-TM4 linker (D230-R242) on the opposing subunit (Fig. 3d). D230 and the SF2-TM4 linker flip upward and outward in a similar way, though to a lesser

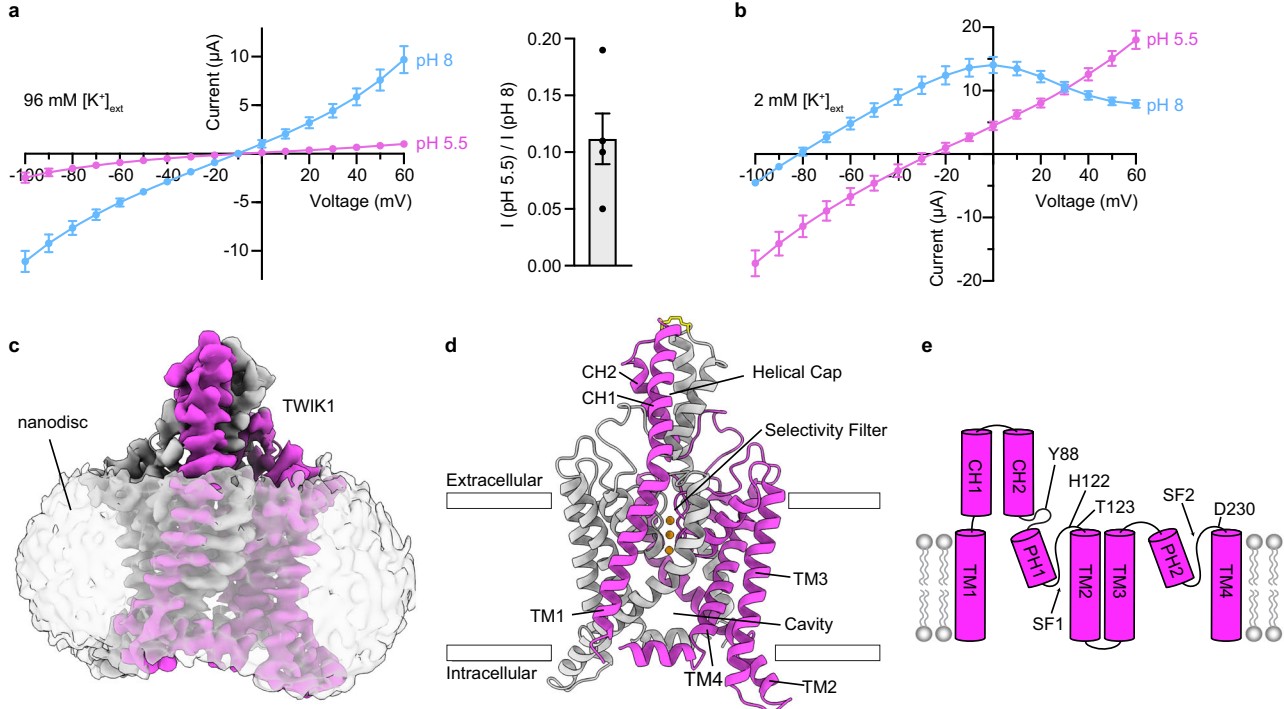

**Fig. 1 Structure and function of TWIK1. a**, **b** Current-voltage relationships from TWIK1-expressing cells at high and low $pH_{ext}$ in (**a**) high $[K^+]_{ext}$ and (**b**) low $[K^+]_{ext}$. **a**, ((right) Fraction of current remaining after extracellular acidification in high $[K^+]_{ext}$ ($I_{pH\,5.5}/I_{pH\,8}$ at 60 mV = 0.11 ± 0.03, mean ± sem from $n = 4$ and 6 cells for high $[K^+]_{ext}$ and low $[K^+]_{ext}$, respectively. **c** Cryo-EM map at pH 5.5 viewed from the membrane plane. Nanodisc is transparent and TWIK1 subunits are magenta and white. **d** TWIK1 model colored as in **c** with $K^+$ ions orange and disulfide yellow. **e** Cartoon representation of a TWIK1 protomer with four transmembrane helices (TM1-TM4), two extracellular cap helices (CH1-CH2), two-pore helices (PH1–PH2), two selectivity filters (SF1–SF2), and key residues involved in extracellular pH-gating indicated. Source data for **a**, **b** are provided as a Source Data file.

degree, positioning D230 ~3.1 Å from protonated H122 (Fig. 3d). Movement of the four linkers is likely concerted because any one linker in a pH 5.5 position sterically clashes with a neighboring linker in a pH 7.4 position. Viewed from above, the linkers push one another into pH 5.5 positions like four dominos falling counterclockwise and form a new set of interactions (Fig. 3d). D230s form salt bridges with H122s on either side of an interaction between opposing T123s (Fig. 3d). These three amino acids from each protomer form a zipper at low pH directly above the selectivity filter to block ion conduction and stabilize the nonconductive conformation (Fig. 3d) (Supplemental Movie 1).

To test this structural model for $pH_{ext}$ gating in TWIK1, we mutated residues implicated in pH-gating and recorded channel activity in whole cell voltage clamp recordings in high $[K^+]_{ext}$ and varying $pH_{ext}$. Wild-type TWIK1 is ~90% inhibited at pH 5.5 relative to pH 8 (Figs. 1a and 3h). Consistent with previous reports and its role as a proton sensor[4], mutation of H122N abolished pH sensitivity (Fig. 3h). While T123 is integral to the pH gate, we predicted mutations at this site would have little effect on inhibition. This is because association of T123s at the center of the zipper above the selectivity filter at pH 5.5 is mediated by interchain peptide backbone interactions while the threonine side chains are exposed to solution in both conformations (Fig. 3b–d). Indeed, mutants at this site showed indistinguishable (T123V, T123G, and T123D) or modestly compromised (T123C) low pH inhibition relative to the wild-type channel (Fig. 3h). In contrast, replacing the acidic D230 with a hydrophobic group in D230L, which we predicted would disfavor the low pH conformation due to its proximity to protonated H122, eliminates pH inhibition (Fig. 3h). A more conservative mutation at this position, D230N, was inhibited to a degree comparable to wild-type TWIK1 (Fig. 3h). Retention and loss of

$pH_{ext}$ sensitivity in D230N and H122N mutants, respectively, further supports the role of H122 as the proton sensing extracellular residue. We note that while all channels analyzed were active, $K^+$ vs. $Na^+$ selectivity was reduced in some cases judged by shifts in reversal potential in low $[K^+]_{ext}$ solution (Supplementary Fig. 5). This is consistent with previous work showing that mutations in these amino acid positions (or interacting positions) alter selectivity in other channels[25,26] and may be due to changes in selectivity filter structure. We conclude pH-gating critically involves residues in both SF1-TM2 and SF2-TM4 linkers to sense protons and seal the channel gate.

Movement of the SF1-TM2 and SF2-TM4 linkers upon protonation of H122 results in two additional large-scale conformational changes. First, the entire helical cap is displaced upward ~4 Å at low pH (Fig. 4a, b, Supplemental Movie 2). Upward movement of the cap is a necessary result of H122 and the SF1-TM2 linker flipping up above the selectivity filter (Fig. 4c). H122-L126 in its pH 5.5 position would clash with E84-G89 at the bottom of CH1 in its pH 7.4 position (Fig. 4c). The essentially rigid body movement of CH2 and the outer half of CH1 (r.m.s.d. of L56-N95 = 0.8 Å) is made possible by P47, which kinks the otherwise continuous helix that forms TM1 and CH1 (Fig. 4c, Supplementary Fig. 6a). The kink straightens at low pH by ~12°, lifting CH1 and pushing TM1 inwards towards PH1 (Fig. 4c, Supplementary Fig. 6a). PH1 then bends up at its extracellular end and creates enough slack in the linker connected to CH2 for the cap to rise (Fig. 4c, Supplementary Fig. 6a). The correlated cap, SF1-TM2 linker, and SF2-TM4 linker movements seal the extracellular ion access pathways to the mouth of the pore (Fig. 2d). At high pH, a wide-open path ~3 Å in radius leads from the extracellular solution to the selectivity filter above H122 and D230 and below CH2s (Fig. 2d). At low pH, the rearranged

**Table 1 Cryo-EM data collection, refinement, and validation statistics.**

| | TWIK1 pH 7.4 EMDB 25168 PDB 7SK0 | TWIK1 pH 5.5 EMDB 25169 PDB 7SK1 |
|---|---|---|
| **Data collection and processing** | | |
| Magnification | 36,000 x | 36,000 x |
| Voltage (kV) | 200 | 200 |
| Micrographs (no.) | 5014 | 3217 |
| Electron exposure (e⁻/Å²) | 49.715 | 50.3523 |
| Defocus range (µm) | −0.6 to −2.0 | −0.6 to −2.0 |
| Super-resolution pixel size (Å) | 0.5685 | 0.5575 |
| Map pixel size (Å) | 1.137 | 1.115 |
| Symmetry imposed | C2 | C2 |
| Topaz particle images (no.) | 2,872,868 | 8,990,305 |
| Final particle images (no.) | 92,438 | 38,126 |
| Map resolution (Å) | 3.4 | 3.4 |
| FSC threshold | 0.143 | 0.143 |
| **Refinement** | | |
| Model resolution (Å) | 3.4 | 3.4 |
| FSC threshold | 0.143 | 0.143 |
| Map sharpening B factor (Å2) | −112.7 | −111.91 |
| Model composition | | |
| Nonhydrogen atoms | 4,153 | 4,297 |
| Protein residues | 514 | 524 |
| Ligands | 7 | 7 |
| B factors (Å²) | | |
| Protein | 147.3 | 143.13 |
| Ligand | 78.45 | 108.96 |
| R.m.s. deviations | | |
| Bond lengths (Å) | 0.007 | 0.008 |
| Bond angles (°) | 1.101 | 1.067 |
| **Validation** | | |
| MolProbity score | 1.14 | 0.99 |
| Clashscore | 1.2 | 0.7 |
| Poor rotamers (%) | 0 | 0.43 |
| Ramachandran plot | | |
| Favored (%) | 95.69 | 96.15 |
| Allowed (%) | 4.31 | 3.85 |
| Disallowed (%) | 0 | 0 |

linkers form the base of a wall topped by the displaced helical cap to block ion access to the filter (Fig. 2d). While the helical cap is involved in binding inhibitory cations in TREK1 (K2P2.1) and TASK3 (K2P9.1) within extracellular ion access pathways[27–29], and TRAAK (K2P4.1) crystal structures display bending of the helical cap off of the two-fold symmetry axis[23,24,30], movement of the helical cap has not previously demonstrated to contribute to K2P gating.

Second, movements of TWIK1 at low pH are propagated to the intracellular side of the membrane to dilate a lateral membrane opening, permitting acyl chain binding in the channel cavity to block conduction (Supplementary Fig. 7). TM2 rocks ~12° so that its intracellular half moves down towards the cytoplasm ~2–4 Å (Supplementary Fig. 6b). This is due to the top of TM2 being pulled (~3 Å by the SF1-TM2 linker rearrangement) and pushed (by straightening of TM1) towards the conduction axis (Supplementary Fig. 6b). TM2 movement propagates to TM4, which tilts downwards by 4° and 1–3 Å (Supplementary Fig. 6c). The resulting rearrangement of amino acids on TM2 and TM4, including I142, L146, M260, and L264, widens the lateral membrane opening from a cross-sectional area of 11.8 Å² at pH 7.4–22.6 Å² at pH 5.5 at its most constricted point (Supplementary Fig. 7a–c). At low pH, clear tube-shaped density consistent with a lipid acyl chain is observed extending from the dilated lateral membrane opening to just under the selectivity filter where it would sterically block ion conduction

(Supplementary Fig. 7c, f, g). The constriction in the lateral membrane opening at high pH is slightly smaller than the cross-section of an acyl-chain methylene (12.6 Å) and, consistently, similar acyl chain density is not observed in the TWIK cavity at high pH (Supplementary Fig. 7d,h). At low pH, weak density extends from the acyl chain into the lipid nanodisc and down towards the intracellular leaflet, but it is insufficiently featured to accurately model a head group, suggesting a lack of specific interaction sites with the channel. While the acyl chain may be contributed by a residual detergent molecule from TWIK1 purification, we reason it is more likely contributed by a lipid molecule for the following reasons: (1) purification and nanodisc reconstitution took place under conditions of high pH that favor low acyl chain occupancy prior to a final chromatography step at low pH without detergent, (2) all three lipid species used in the nanodisc reconstitution could reside in a low energy conformation with a headgroup in the inner leaflet and an acyl chain bound in the channel cavity as observed based on geometric and steric considerations, and (3) lipid acyl chains are present at higher concentrations in the nanodisc relative to potential residual detergent and would likely have higher fractional occupancy in the loose hydrophobic binding site of the channel cavity.

The TWIK1 lateral membrane opening at high pH is larger than that observed in other conductive K2P structures, including TRAAK and TASK2 (Supplementary Fig. 7h), and just small enough to prevent lipid acyl chain access to the cavity. While the conformation we observe at high pH is likely predominant under these conditions, we cannot exclude the existence of a subpopulation of particles with a slightly larger opening containing cavity bound acyl chains. The relatively large lateral membrane opening at high pH may contribute to low activity of TWIK1 in cells relative to other K2Ps[3] because only subtle rearrangements would expand it sufficiently for acyl chain binding and channel block. An alternative explanation based on molecular dynamics simulations is that TWIK is prone to pore dewetting[16–18]. The crystal structure of human TWIK1 at pH 8 showed a wide lateral membrane opening, cavity bound acyl chains blocking the conduction path, and TM2, TM4, and the helical cap positioned closer to the low pH than high pH structure reported here (Supplementary Fig. 8)[9]. What could account for the differences between the high pH structure reported here and the crystal structure? One possibility is that TWIK1 samples a lipid-blocked closed conformation at high pH that was captured by crystal contacts. Alternatively, different constructs, solutions, or solubilizing environments (lipid nanodisc versus detergent micelle) could promote different TWIK1 conformations.

## Discussion

The physiological relevance of lipid block of K2Ps in the channel cavity is debated. Structural evidence of acyl chain binding in the cavity has been reported for K2Ps in detergent micelles (TRAAK[10], TREK2[11], and TWIK1[9]) and lipid nanodiscs (TASK2[12] and TWIK1). A model for TRAAK gating from our group posits that lipid block underlies a long duration TM4 down closed state[31]. In contrast, though acyl chain binding in the cavity of TREK2 is observed structurally[11] and in molecular dynamics simulations[32], a model for TREK2 gating based on these and other data does not involve lipid block[32,33]. Molecular dynamics simulations of TWIK1 based on the previously reported crystal structure showed acyl chain binding in the cavity, but it was concluded that lipids did not directly block conduction and rather influenced spontaneous dewetting of the pore[16–18]. Mutations that increase hydrophilicity of a conserved position facing the interior of K2P cavities increases activity across the family[5,34]. We concluded based on structural and functional data that the

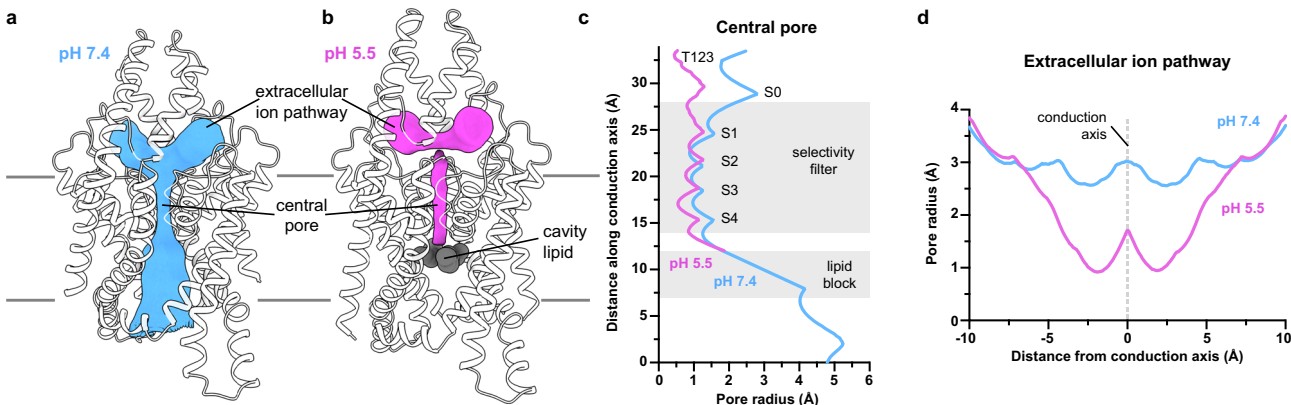

**Fig. 2 TWIK1 conduction pathways at high and low pH. a**, **b** Structures of TWIK1 at (**a**) pH 7.4 and (**b**) pH 5.5 with the surface of conduction pathways shown in gray. **c** Radius of the central pore as a function of distance along the conduction pathway. Positions of the selectivity filter, lipid block at low pH, K$^+$ coordination sites S0-S4, and T123 are indicated. **d** Radius of the extracellular ion pathways as a function of distance from the central conduction axis.

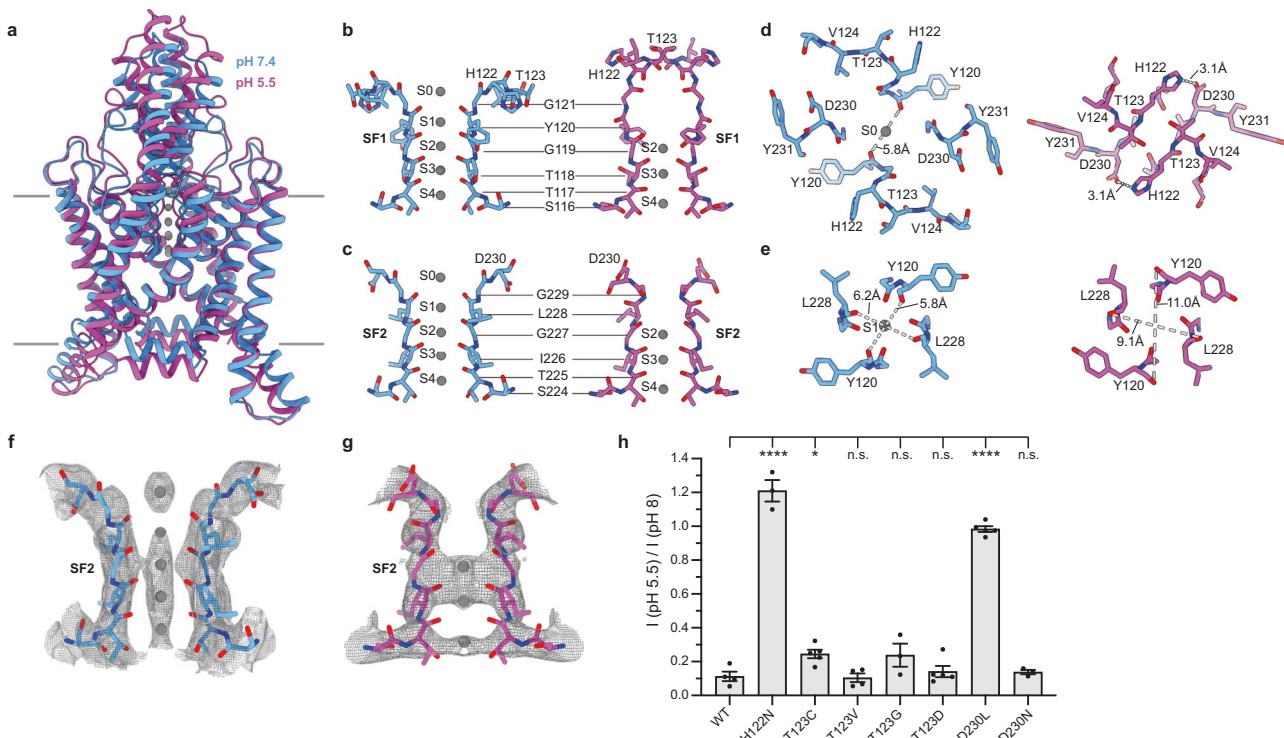

**Fig. 3 TWIK1 selectivity filter gate controlled by pH$_{ext}$. a** Overlay of open (pH 7.4, blue) and closed (pH 5.5, magenta) conformations of TWIK1 viewed from the membrane plane. **b**–**e** Comparisons of regions involved in the selectivity filter gate from open (left) and closed (right) TWIK1 structures. **b** SF1, **c** SF2, **d** the top of the selectivity filter, and **e** coordination site S1 are shown **b**, **c** from the membrane plane and **d**, **e** from the extracellular side. K$^+$ coordination sites S0-S4 and intra-carbonyl or H122-D230 salt bridge distances are indicated. **f**, **g** Cryo-EM maps around the selectivity filter illustrating ion occupancy in **f** open and **g** closed structures. **h** Fraction of current remaining upon extracellular acidification (pH$_{ext}$ = 5.5/pH$_{ext}$ = 8.0 at 60 mV) for wild-type TWIK1 (0.11 ± 0.03) and mutants H122N (1.21 ± 0.06), T123C (0.24 ± 0.02), T123V (0.10 ± 0.02), T123G (0.24 ± 0.07), T123D (0.14 ± 0.03), D230L (0.98 ± 0.02), and D230N (0.14 ± 0.01). Mean plus s.e.m. were plotted for $n = 4, 3, 5, 4, 3, 5, 5$, and 3 cells, respectively. Differences were assessed with one-way analysis of variance (ANOVA) with Dunnett correction for multiple comparisons (*$P = 0.0425$; ****$P < 0.0001$; n.s. not significant). Source data for **h** are provided as a Source Data file.

mechanism for increased activity of TRAAK G158D is elimination of lipid block[31]. In contrast, others concluded the mechanism for activation of TWIK1 L146N/L146D was rather a reduction in pore dewetting[18]. Further study is necessary to determine the relevance of direct lipid gating and the degree of interaction between the extracellular gate at the selectivity filter and lipid block through the lateral membrane opening in TWIK1.

Selectivity filter gating (sometimes referred to as C-type gating) has been long appreciated to play a central role in the function of

many K$^+$ channels[35], but its structural basis is just beginning to be understood. Here, we demonstrate TWIK1 utilizes a previously unobserved type of selectivity filter gate in which concerted movement of filter-adjacent linkers disrupts K$^+$ coordination sites S0 and S1 and zippers shut the mouth of the channel pore. Still, one aspect of this mechanism, the upward movement of H122 and D230 above the selectivity filter in response to low pH$_{ext}$, is reminiscent of upward movements by analogous residues in the K2P TREK1[36], the bacterial K$^+$ channel KcsA[37], and the voltage-gated

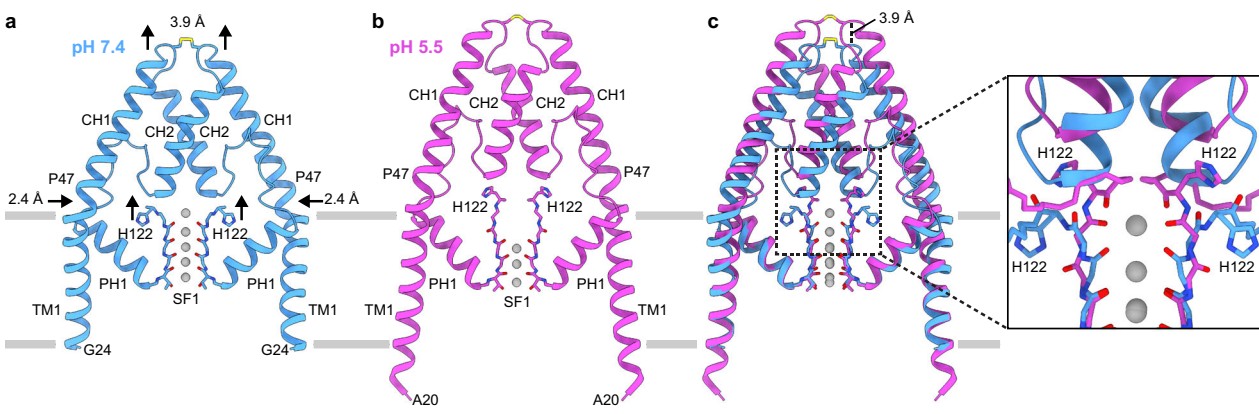

**Fig. 4 Helical cap and transmembrane helix rearrangements in response to pH$_{ext}$. a–c** View of TM1s, helical cap, PH1s, and SF1s at (**a**) pH 7.4, (**b**) pH 5.5, and (**c**) overlaid. Arrows highlight the upward movements of H122 and helical cap and the inward movement of TM1 in response to low pH$_{ext}$. **c** (inset) Zoomed in view of region boxed in **c** illustrating steric overlap between the selectivity filter in the low pH$_{ext}$ structure and helical cap in the high pH$_{ext}$ structure.

K$^+$ channel Shaker[38] (Supplementary Figs. 9 and 10). In a KcsA E71A mutant, D80 flips upward and is exposed to the extracellular solution like TWIK1 H122, but this conformational change does not block ion conduction (Supplementary Fig. 9b, c)[37] and may be promoted by interactions with an antibody Fab fragment used to crystallize the channel. In TREK1, N147, which is analogous to H122 of TWIK1, flips up in response to changes in [K$^+$]$_{ext}$. While D256 of TREK1, which is analogous to D230 of TWIK1, also flips up in response to low [K$^+$]$_{ext}$ (Supplementary Fig. 9a), SF2 of TREK1 dilates in a manner distinct from the conformational rearrangements reported here for TWIK1[36]. A similar upwards movement of this conserved aspartic acid (D447) was recently observed in a rapidly inactivating mutant of Shaker (Supplementary Fig. 9d, e)[38]. The proton sensor H122 is conserved in TRESK (Supplementary Fig. 10) and pH-sensitive TASK subfamily K2Ps, and TASK1 and TASK3 have been reported to lose selectivity when exposed to low pH$_{ext}$ like TWIK1[7]. Whether conformational changes similar to those reported here are involved in TASK channel gating and whether [K$^+$]$_{ext}$-dependent changes in selectivity are conserved between TWIK and TASK channels remains to be determined.

Other structurally characterized selectivity filter gates are more distinct from what we observe in TWIK1. The TALK-subfamily K2P TASK2, like TWIK1, is inhibited by external protons through a filter gate, but the molecular mechanisms involved are unrelated. The extracellular proton sensor of TASK2 is located further from the pore near the top of TM4 and TASK2 conformational changes at the filter are more subtle, involving rearrangement of coordination sites S0 and S1 that render the channel nonconductive[12]. Finally, C-type gating at the selectivity filter in an inactivating mutant of a chimeric Kv1.2-2.1 channel involves only constriction of carbonyls around S1 to disrupt K$^+$ coordination[39]. The discovery of a previously undescribed gating mechanism in TWIK1 adds to our understanding of the myriad ways in which the selectivity filter can gate K$^+$ channels to control their activity.

The structures reported here support a model for TWIK1 gating by pH$_{ext}$ under conditions of high [K$^+$]$_{ext}$ (Fig. 5). Under low pH, low [K$^+$]$_{ext}$ conditions, TWIK1 is expected to sample a different open conformation with reduced K$^+$ selectivity that underlies its intriguing reversible shift in relative ion permeability. What aspects of the selectivity filter structure might promote a less K$^+$ selective conformation in TWIK1? Reduced structural integrity of the TWIK1 filter relative to other K$^+$ channels has been suggested to be an important factor because a unique

hydrophilic residue in the first selectivity filter (T118) is critical for selectivity changes[5] and the selectivity filter is only weakly stabilized by K$^+$ in a conductive conformation[20]. We observe a large cavity (~78 Å$^3$) behind the selectivity filter in the closed structure reported here created by movement of H122 and T123 that may contribute to reduced filter stability under these conditions. Future studies of TWIK1 at low [K$^+$]$_{ext}$ will reveal how other structural rearrangements account for increased Na$^+$ permeability.

## Methods

**Cloning, expression, and purification.** Cloning, expression, and purification were performed similarly to that described for the K2P channel TRAAK[10]. A gene encoding *Rattus norvegicus* TWIK1 (Uniprot Q9Z2T2) was codon-optimized for expression in *Pichia pastoris*, synthesized (Genewiz, Inc), and cloned into a modified pPICZ-B vector (Life Technologies Inc). The resulting construct encoded a human rhinovirus 3 C protease-cleavable C-terminal EGFP-10x histidine fusion protein. The resulting construct, TWIK1$_{-SNS-LEVLFQ/GP-(EGFP)-HHHHHHHHHH}$ was used for structural studies and is referred to as TWIK1 in the text for simplicity.

Pme1 linearized pPICZ-B plasmid was transformed into *Pichia pastoris* strain SMD1163 by electroporation and transformants were selected on YPDS plates with 1 mg/ml zeocin. Expression levels of individual transformants were analyzed by florescence size. For large-scale expression, overnight cultures of cells in YPD + 0.5 mg/mL Zeocin were added to BMGY to a starting OD of 1, and grown overnight at 30 °C to a final OD$_{600}$ of 25. Cells were centrifuged at 8000 x g for 10 min, and added to BMMY at 27 °C to induce protein expression. Expression continued for ~24 h.

Cells were pelleted, flash-frozen in liquid nitrogen, and stored at −80 °C. 60 g of cells were broken by milling (Retsch model MM301) for 5 cycles of 3 minutes at 25 Hz. All subsequent purification steps were carried out at 4 °C. Cell powder was added to 200 mL lysis/extraction buffer (50 mM Tris pH 8.0, 150 mM KCl, 1 mM phenylmethysulfonyl fluoride, 1 mM EDTA, 10 µl Benzonase Nuclease (EMD Millipore), 1 µM AEBSF, 1 mM E64, 1 mg/ml Pepstatin A, 10 mg/ml Soy Trypsin Inhibitor, 1 mM Benzimidine, 1 mg/ml Aprotinin, 1 mg/ml Leupeptin, 1% n-Dodecyl-b-D-Maltopyranoside (DDM, Anatrace, Maumee, OH), 0.2% Cholesterol Hemisuccinate Tris Salt (CHS, Anatrace)) and then gently stirred at 4 °C for 3 h. The extraction was then centrifuged at 33,000 × g for 45 minutes. 10 ml Sepharose resin coupled to anti-GFP nanobody was added to the supernatant and stirred gently for 1 h at 4 °C. The resin was collected in a column and washed with 50 mL Buffer 1 (20 mM Tris, 150 mM KCl, 1 mM EDTA, 1% DDM, 0.2% CHS, pH 8.0), 150 mL Buffer 2 (20 mM Tris, 300 mM KCl, 1 mM EDTA, 1% DDM, 0.2% CHS, pH 8.0), and 100 mL of Buffer 1. PPX (~0.5 mg) was added into the washed resin in 10 mL Buffer 1 and rocked gently overnight. Cleaved TWIK1 was eluted and concentrated to ~9.6 mL with an Amicon Ultra spin concentrator (30 kDa cutoff, MilliporeSigma, USA). The concentrated protein was subjected to size exclusion chromatography using a Superdex 200 Increase 10/300 column (GE Healthcare, Chicago, IL) run in Buffer 3 (20 mM Tris pH 8.0, 150 mM KCl, 1 mM EDTA, 1% DDM, 0.01% CHS) on a NGC system (Bio-Rad, Hercules, CA) running ChromLab 6.0. The peak fractions were collected and spin concentrated for reconstitution.

**Nanodisc formation.** 10 nmol of freshly purified TWIK1 was reconstituted into nanodiscs with a 2:1:1 DOPE:POPS:POPC lipid mixture (mol:mol, Avanti, Alabaster,

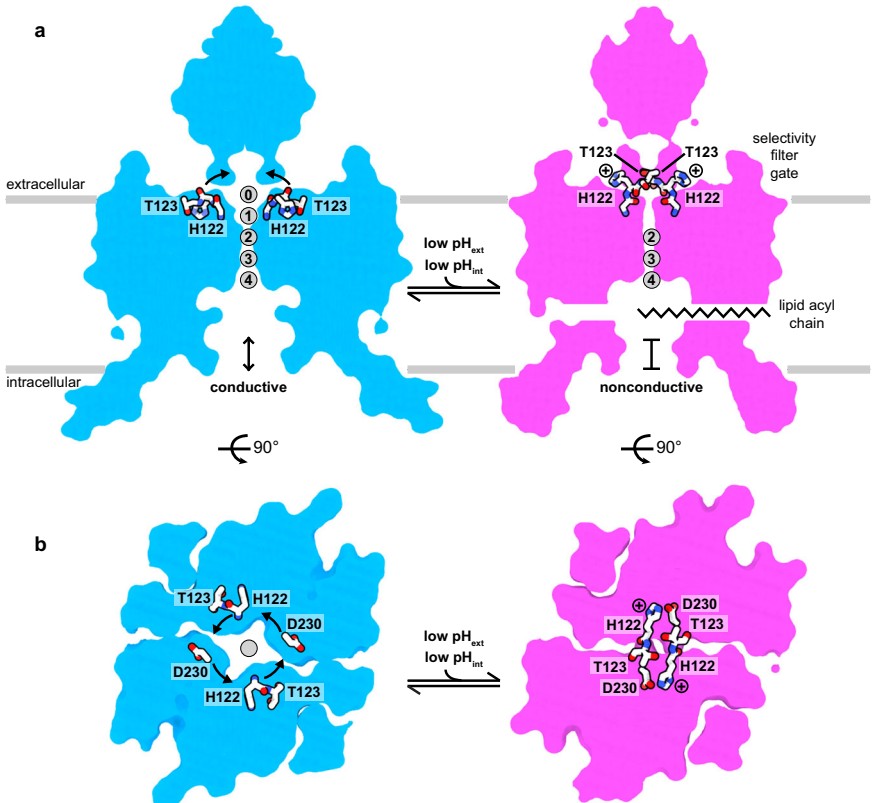

**Fig. 5 Structural model for pH-gating of the TWIK1 channel. a, b** Model for selectivity filter gating of TWIK1 gating by pH_ext viewed from (**a**) the membrane plane and (**b**) the extracellular side. TWIK1 is conductive at a high pH. At low pH, H122 protonation results in conformational changes that disrupt K⁺ coordination sites S0 and S1, seal the top of the selectivity filter, displace the helical cap to block extracellular ion access pathways, and dilate a lateral membrane opening to permit acyl chain binding to the channel cavity.

Alabama) at a final molar ratio of TWIK1:MSP1D1:lipid of 1:5:250 for high pH, or a ratio of TWIK1:MSPE3D1:lipid of 1:2:200 for low pH. Lipids in chloroform were mixed, dried under argon, washed with pentane, dried under argon, and dried under vacuum overnight in the dark. Dried lipids were rehydrated in buffer containing 20 mM Tris, 150 mM KCl, 1 mM EDTA, pH 8.0 and clarified by bath sonication. DDM was added to a final concentration of 8 mM. TWIK1 was mixed with lipids and incubated at 4 °C for 1 h before addition MSPE3D1 protein. After incubation for 30 minutes at 4 °C, 100 mg of Biobeads SM2 (Bio-Rad, USA) (prepared by sequential washing in methanol, water, and Buffer 4 and weighed damp following bulk liquid removal) was added and the mixture was rotated at 4 °C overnight. The sample was spun down to facilitate removal of solution from the Biobeads and the reconstituted channel was further purified on a Superdex 200 increase column run in 20 mM HEPES pH 7.4, 150 mM KCl, and 1 mM EDTA (high pH) or 20 mM MES pH 5.5, 150 mM KCl, and 1 mM EDTA (low pH). The peak fractions were collected and spin concentrated (50 kDa MWCO) to 1.8 mg/mL (for high pH) or 2.5 mg/ml (for low pH) for grid preparation.

**Grid preparation.** The TWIK1 nanodisc samples were centrifuged at 21,000 x g for 10 min at 4 °C. A 3 μL sample was applied to holey carbon, 300 mesh R1.2/1.3 gold grids (Quantifoil, Großlöbichau, Germany) that were freshly glow discharged for 30 seconds. Sample was incubated for 5 seconds at 4 °C and 100% humidity prior to blotting with Whatman #1 filter paper for 3 seconds at blot force 1 and plunge-freezing in liquid ethane cooled by liquid nitrogen using a FEI Mark IV Vitrobot (FEI/Thermo Scientific, USA).

**Cryo-EM data acquisition.** Grids were clipped and transferred to a FEI Talos Arctica electron microscope operated at 200 kV. Fifty frame movies were recorded on a Gatan K3 Summit direct electron detector in super-resolution counting mode with pixel size of 0.5575 Å. The electron dose was 9.039 e⁻ Å² s⁻¹ and 9.155 e⁻ Å² s⁻ and total dose was 49.715 e⁻ Å² and 50.3523 e⁻ Å² in pH 7.4 and pH 5.5 datasets, respectively. Nine movies were collected around a central hole position with image shift and defocus was varied from −0.8 to −2.0 μm through SerialEM[40]. See Table 1 for data collection statistics.

**Cryo-EM data processing.** For TWIK1 in nanodiscs at pH 7.4, 3,652 micrographs were corrected for beam-induced drift using MotionCor2 in Relion 3.1 and the data was binned to 1.115 A/pixel[41–43]. The contrast transfer function (CTF) parameters

for each micrograph were determined using CTFFIND-4.1[44]. For particle picking in TWIK1 at high pH, 1000 particles were picked manually and subjected to reference-free 2D classification in RELION 3.1 to generate reference for autopicking. After initial cleanup through rounds of 2D classification in Relion 3.1, the remaining particles were extracted and imported into cryoSPARC[45]. CryoSPARC was used to generate an ab initio model with 2 classes and 0 similarity with or without symmetry. Particles belonging to a class with well-defined features were further refined using Homogenous refinement.

Particle positions and angles from the final cryoSPARC2 refinement job were input into Relion 3.1 (using csparc2relion.py from the UCSF PyEM[46]), subjected to 3D classification (4 classes, tau 4, 7.5 degrees global sampling), and refined again to produce a 4.5 Å map. Bayesian polishing, refinement, an additional round of 3D classification (4 classes, tau 16, no angular sampling), and 3D refinement improved the map further to 3.7 Å. CtfRefinement and Particle Subtraction led to a map at 3.6 Å. Particles were then repicked with Topaz[47], and subjected to a processing procedure similar to what is described above, leading to a map at 3.4 Å. Particles leading to this map were merged with particles leading to the best map generated before repicking with Topaz, and duplicates were removed, leading to a map at 3.5 Å. This one had less anisotropy than the one produced without Topaz, and allowed for clearer visualization of the C-terminal helix. An additional round of CTF refinement, particle subtraction to remove the contribution of the nanodisc density and subsequent 3D refinement yielded a map at 3.4 Å and used for model building.

For TWIK1 in nanodiscs at pH 5.5, 7,859 micrographs were collected. The topaz model for the high pH data set described above was used to extract ~9.6 million particles, and subjected to a data processing procedure that was similar as above, except that in this case, 2D classification was done exclusively in cryoSPARC. Bayesian particle polishing, CTF refinement, and particle subtraction yielded a final map at 3.4 Å.

**Modeling and refinement.** Cryo-EM maps were sharpened using Relion LocalRes[48] in the case of TWIK1 at low pH or Phenix Density Modification in the case of TWIK1 at high pH[49]. The initial model was built from PDB 3UKM (https://www.rcsb.org/structure/3UKM)[9] and refined into the density for TWIK1 at high pH using Phenix.real_space_refine with Ramachandran and NCS restraints[50]. In the high pH structure, the bottom of TM2 and the TM2-TM3 linker (T161-A180) are much lower in resolution, possibly due to potential conformational flexibility.

For this region, the model was first built in the low pH structure, using a model predicted by AlphaFold2[51], refined, and then docked into the high pH structure. Molprobity[52] was used to evaluate the stereochemistry and geometry of the structure for manual adjustment in Coot[53] and refinement in Phenix. Cavity measurements were made with HOLE implemented in Coot[54]. Figures were prepared using PyMOL, Chimera, ChimeraX, and Prism.

**Electrophysiology**. Full-length TWIK1 was cloned into a modified pGEMHE vector using Xho1 and EcoR1 restriction sites such that the mRNA transcript encodes full-length TWIK1 with an additional "SNS" at the C-terminus. Three mutations (K274E, I293A, I294A) were introduced into this construct via inverse PCR to improve trafficking of the channel to the cell membrane, remove any possibility of SUMOylation, and increase basal currents. This construct was used as the background construct for all other mutants in this study, which were also generated via inverse PCR. The following primers were used: I923A/M294Afwd CATGACCAACTGAGCTTC, I923A/M294Arev AACTTGATCCTCATCCTTGT, H122Nfwd CCATTGTCGGATGGT, H122Nrev ACCGGTAGTGGACAA, T123Cfwd TTGTCGGATGGTGGAAAG, T123Crev GTAACCGGTAGTGGAC AA, T123Vfwd TTGTCGGATGGTGGAAAG, T123Vrev GTAACCGGTAGTGG ACAA, T123Gfwd TTGTCGGATGGTGGAAAG, T123Grev GTAACCGGTA GTGGACAA, T123Dfwd TTGTCGGATGGTGGAAAG, T123Drev GTAACCG GTAGTGGACAA, D230Lfwd CCTGGAGAGGGCTACAAC, D230Lrev GCC GATAGTGCGAGAGTGA, D230Nfwd CCTGGAGAGGGCTACAAC, and D230 Nrev GCCGATAGTGCGAGAGTGA.

cRNA was transcribed from Nhe1-linearized plasmids in vitro using T7 mMessage mMachine kits, and 2.5–3.0 ng cRNA was injected into Stage V–VI *Xenopus laevis* oocytes extracted from anaesthetized frogs. Currents were recorded at 25 °C using two-electrode voltage clamp (TEVC) from oocytes 2 days after mRNA injection. Pipette solution contained 3 M KCl. Bath solution contained either ND96 (96 mM NaCl, 2 mM KCl, 1.8 mM CaCl$_2$, 1 mM MgCl$_2$, 2.5 mM Na pyruvate, 20 mM buffer), or ND96K (2 mM NaCl, 96 mM KCl, 1.8 mM CaCl$_2$, 1 mM MgCl$_2$, 2.5 mM Na pyruvate, 20 mM buffer) at a pH of 8, or 5.5. HEPES was used to buffer each solution to a pH of 8, and MES was to buffer to pH 5.5. Bath volume was kept at 300 uL during recording. Eggs were initially placed in the recording chamber in ND96, with ND96K at pH 8 perfused in the bath afterwards, followed by ND96K at pH 5.5. Currents were recorded and low-pass filtered at 2 kHz using a Dagan TEV-200A amplifier and digitized at 10 kHz with a Sutter Dendrite digitizer.

**Reporting summary**. Further information on research design is available in the Nature Research Reporting Summary linked to this article.

## Data availability
The final cryo-EM maps of *R. norvegicus* TWIK1 (Uniprot Q9Z2T2,) in MSP1D1 nanodiscs at pH 7.4 and in MSPE3D1 nanodiscs at pH 5.5 generated in this study have been deposited in the Electron Microscopy Data Bank under accession codes EMD-25168 and EMD-25169. The final atomic coordinates have been deposited in the PDB under IDs 7SK0 and 7SK1. The original micrograph movies have been deposited in EMPIAR. Source Data are available with the paper.

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

## Acknowledgements

We thank Dr. Jonathan Remis, Dr. Dan Toso, and Paul Tobias at UC Berkeley for assistance with microscope setup and data collection. S.G.B. is a New York Stem Cell Foundation-Robertson Neuroscience Investigator. This work was supported by the New York Stem Cell Foundation, NIGMS grant DP2GM123496, a McKnight Foundation Scholar Award, a Klingenstein-Simons Foundation Fellowship Award, and a Sloan Research Fellowship to SGB; as well as an NSF-GRFP to T.S.T..

## Author contributions

T.S.T. performed all molecular biology, biochemistry, cryo-EM, and electrophysiology experiments. V.L. performed preliminary cloning and electrophysiology. T.S.T. and S.G.B. collected cryo-EM data, modeled structures, analyzed data, and wrote the manuscript. S.G.B. supervised the project.

## Competing interests

The authors declare no competing interests.
