## [Peer Review File · Nature Communications]

Structural Basis for pH-Gating of the K⁺ Channel TWIK1 at the Selectivity FilterREVIEWER COMMENTS

Reviewer #1 (Remarks to the Author):

Turney et al. NCOMMS-21-51552-T

This manuscript describes structural studies of the K2P channel K2P1.1 (TWIK1) using cryo-electronmicroscopy. The authors determine structures of TWIK1 in high (pH 8.0) and low (pH 5.5) conditions that should correspond to the active and inactive forms (when the channel is under high potassium concentrations). The structures reveal that the low pH conditions induce a conformational change in the selectivity filter that involves flipping out of key residues, zippering of the filter, loss of ions in the outer sites, constriction of the extracellular ion pathway (EIP) by the upward movement of the cap, and possible block of the ion pathway by lipids that enter the central cavity through lateral caps under the selectivity filter. Together, these changes appear to show another twist on the ways that selectivity filter conformational changes can impact potassium channel function. The general area of 'C-type' gating is now receiving a great deal of attention from a variety of structural studies and as such the results here are timely and important as they further show the remarkable structural plasticity of potassium channel selectivity filters. Nevertheless, the manuscript has a number of key deficits in terms of data quality, presentation, and contextualization and reference to prior studies that need to be addressed.

Major issues:

Two electrode voltage clamp recordings show a surprising variability regarding the reversal potential. These suggest suggest either poor quality oocytes or important selectivity changes. Reversal potentials for WT and the mutations shown in Fig. S5 are varied, ranging from less than zero (WT), zero (T123G, T123D), to substantially above zero (D230N). These changes suggest either that there are substantial alterations to ion selectivity, or that there was poor control of the background leak from the oocytes. The authors even note that the D230N has no effect on pH, implying that this is a wild-type channel. Clearly, if the reversal potential in S5A is correct, D230N is not a wild-type-like channel. Further, in terms of comparison with other published studies in oocytes using similar conditions (ex. Chatelain et al. PNAS 109:5499 (2012)), the WT value appears to be more negative than expected. These issues need to be addressed, especially as there was no comment on these apparent fundamental changes in the properties of the studied channels.

Although the data appear to provide a clear model for how the channel is inhibited by low pH under high potassium conditions, the authors make no attempt to reconcile how this sort of state, which seems to be inhibited three ways 1) closing of the filter, 2) narrowing of the EIP by movement of the cap,

and 3) the proposed block by a cavity lipid, could be relevant for the the dramatic the loss of selectivity seen at low pH / low [K⁺]. In general, the authors do not do a good job at putting the results here in the context of other studies of K2P filter gating, in terms of how this low pH/high[K⁺] inhibited state is or is not related to the ideas of selectivity filter instability (Nematian-Ardestani et al. *J Biol. Chem.* 295:610-618 (2020)), or why the low pH / low [K⁺] state is not only active but allows the passage of sodium ions. As the authors show that their construct undergoes this selectivity loss (Fig. 1B), there should be some sort of discussion on this point. Given the seemingly tightly zippered selectivity filter with additional cavity lipids occluding the filter from below, It does not look like Na⁺ ions would be able to pass through this structure without additional conformational changes - especially in regard to the 'lipid block' which looks like a total occlusion. It seems likely that the leaky state at low [K⁺] and low pH is something quite different than what is observed here.

In the 2nd to last paragraph on page 4 of the pdf which starts "Selectivity filter gating..." they compare results from other studies in regards to how the next residue just outside the filter flips up. The authors write: "In TREK1, N147 flips up in response to changes in [K⁺]_{ext}, although this conformational change is only observed within one subunit and does not fully block the conduction path (Lolicato et al, 2020)." This is a misleading statement. While it is true the N147 flip is only in one subunit in the TREK1 crystal structures, the simulations Lolicato et al., 2020 indicate that this conformational change happens in both subunits (as shown in the symmetrized models in Fig. 6 of Lolicato 2020) and blocks the conduction path. Of note, such interpretations in which changes in one subunit are placed in the context of a symmetric homodimer are in line with prior K2P studies in which movement is observed by crystallography in only one subunit, but symmetrized versions were displayed and used to develop mechanistic models (ex. Brohawn et al. *Nature* 2014). The authors should note both the structural and simulation studies that support a symmetrical change at the N147 site in K2P2.1 (TREK-1).

Both the presentation and discussion of the potential blocking lipid is problematic. The evidence for the lipid is not presented when the concept is first introduced. The block is noted in the cartoon in Fig. 2b, but the data that support this model are in Fig. S7g, which are only introduced much later in the manuscript. Further, from the view in Fig. S7g, it is unclear if the lipid density merges with the density for the ion or is at a different altitude along the channel central axis. The authors also present the lipid block as being equivalent to norfluoxetine. Although the site may be the same, norfluoxetine does not block the inner cavity. This is important distinction needs to be noted as the authors seem to imply common mechanism, where one does not exist.

The question of lipid access to the K2P central cavity remains a controversial one. Tucker et al. have ruled out such mechanisms from both data and simulations (McClenaghan et al. *J. Gen. Physiol.* 147:497-5050 (2016), Aryal et al. *Structure* 25:708-718 (2017)). One cannot exclude from the current data that what the authors see here is the access of residual detergent that enters the cavity in the way that a lipid that is part of the membrane bilayer cannot. The authors are certainly free to offer their own view of their data, but they should address the work of Tucker et al. in the discussion of possible lipid block mechanisms and potential shortcomings. Further, as noted above, the low pH lipid block would

seem to be a major impediment to the sodium flux that happens in the low pH/low [K⁺] state and providing some comment on this seems important.

Statement that the work here first example of the cap being involved in K₂P gating is not true. The cap has been shown to be intimately involved in channel gating by polyruthenium amines and other cations (Gonzalez et al. *J. Biol. Chem.* 228 5984-5991 (2013), Braun et al. *Br. J. Pharmacol.* 172:1728-1738 (2015), Pope et al. *Cell Chem Biol* 27:511-524.e4 (2020)). The movements here have essentially similar structural consequences to modulation by these polycations, namely, physical block of ion access to the EIP. While these structures may be the first example of cap movement being linked to gating, the authors should rephrase their description to acknowledge the prior roles of the cap shown by others and note that there are conceptual links to what they observe here. Accurately expanding this description will improve the generality of their observations and take nothing away from the novel conformational changes they observe here.

Minor points:

The authors should (at least in the abstract and introduction) use the regularized name for this channel, K₂P1.1 (TWIK-1) (and any other K₂P channels that are discussed). The authors should include these names (recognized by the International Union of Basic and Clinical Pharmacology. <http://journals.ed.ac.uk/gtopdb-cite/article/view/6417>) to ensure that readers are clear about which channels are under discussion here. It is also important for various literature searches that this official name is found in the manuscript, even if the authors then choose to revert to 'TWIK1' for the rest of their writing.

In Figs. 3 f & g, the structure is labeled SF1, but based on the amino acids present it appears to be SF2.

When the authors compare the changes here with other C-type gating changes, it would seem appropriate to cite the work for the various examples upon first mention (i.e. in the line 'reminiscent of upward movements by analogous residues in the K₂P TREK1 (ref), the archaeal channel KcsA (ref), (Note: KcsA does not come from an archaeon but from a gram-positive bacterium. Please correct this)...the voltage-gated K⁺ channel Shaker (ref).

There seems to be an error in the third paragraph of the introduction 'add Aryal et al'. It looks like this was a 'note to self' during the drafting of this manuscript and represents an incomplete edit.

References to the relevant papers should be made along with the PDB codes in Figure S9.

The Supplementary movies were not part of the review package. I assume that they are identical to those posted on bioRxiv

(<https://www.biorxiv.org/content/10.1101/2021.11.09.467928v1.supplementary-material>)

If so, the movies are fine and clearly present the motions under discussion.

Reviewer #2 (Remarks to the Author):

TWIK1 is a K2P channels that is highly expressed in the brain and heart and plays an important role in cardiac rhythm generation and insulin release. It is gated by pH, closing at pH 5.5 and opening at pH 7.5. Although a x-ray structure of TWIK1 at high pH has been published, the pH-dependent gating of TWIK1 remains unknown. In this manuscript, Turney et al. determined cryo-EM structures of TWIK1 at high and low pH in nanodisc at resolutions up to 3.4Å. They observed different conformations at different pH – at high pH, TWIK1 is in an open conformation while at low pH it changes to a closed conformation. Comparing these two structures revealed a cascade of conformational changes throughout the protein, providing a gating mechanism of TWIK1 induced by pH. They observed a conformational change in the selectivity filter from high pH to low pH and elucidated an interesting gating mechanism associated with selectivity filter. Since TWIK1 has unique properties in K2Ps, the results presented by the authors provide a mechanistic understanding of its unique function properties. Overall, this is an interesting and important study of K2P channels that will deepen our understanding on the pH-dependent gating mechanism of TWIK1. While the structural and functional studies here are solid, I do have several comments that the authors need to address before the manuscript can be accepted.

1. The manuscript has some readability issues. First, the figures are not properly labeled. For instance, the “helical cap” are mentioned throughout the text, but are not labeled in the figures. The same is true for the “SF2-TM4 linker” or “SF1-TM2 linker” etc. The authors should read through the text and figures to make sure that everything is labeled properly. Second, the Figures are not properly referenced throughout the text. For instance, there are eight panels in Figure 3, but the authors only cited “Figure 3” in the text without citing each panel. This makes it difficult for the readers to connect the text to the figures; Another example is the paragraph on page 2 “At high pH, the selectivity filter adopts a nearly four-fold symmetric ...”, where no figure was cited even though extensive conformational changes were described. Third, a considerable amount of conformational changes described in the text are not shown (or can’t be found) in the figures. For instance, “H122 flips upward towards the helical cap. The adjoining linker region connecting SF1 to TM2...”. Overall, the authors should have done a better job of improving readability.

2. It is unclear how can the clash happen in “In its pH 5.5 position, H122-L126 clashes with E84-G89 at the bottom of EC2”, and how it is linked to the gating mechanism. Please clarify.
3. What is the gray molecule in Figure S7c?
4. Why the crystal structure of TWIK1 at high pH adopted the same conformation at low pH?

Reviewer #3 (Remarks to the Author):

K2P potassium channels provide the background conductance of all cells. They are exquisitely regulated by a variety of stimuli and mutations affecting them are the cause of several human diseases. TWIK-1 is the founder member of the K2P family and has been extensively studied. TWIK-1 is gated by pH through a known pH sensor, an extracellular facing histidine residue, but details of the molecular mechanism involved are not known. More generally, the molecular mechanisms of gating of the K2P channels is only beginning to be understood.

This paper addresses the mechanism of gating of TWIK-1 K2P potassium channel by pH through the resolution by cryo electron microscopy of the structures of TWIK-1 in a lipid environment at neutral and acidic pH, which correspond to conditions under which the channel is open and closed respectively.

The structures reveal a complex mechanism of gating. The channel exhibits a continuous open permeation pathway at neutral pH while at acid pH a change in orientation of the sensing histidine is accompanied by alterations at the extracellular access to the pore and the upper reaches of the selectivity filter prohibiting potassium flow. Remarkably, the closed state is also associated with intramembrane fenestration changes allowing lipid entry and further inner pore blockade of the permeation pathway.

The results give important novel ideas of the mechanisms of gating in K2P channels. They offer the first molecular description of widespread C-type inactivation in K2P channels. It also reaffirms the importance of lipid intrusion in gating and provides a function for intramembrane fenestrations in K2P channels as gated structures.

The work is original and clearly of high significance for our understanding of K2P channel regulation. The methodology employed is sound and the authors' laboratory are leaders in the field. The presented

evidence fully supports the contentions of the paper and there no caveats in the data analysis. The paper is succinct but data are well presented.

A minor point concerns a problem with the presentation of functional results of the effect of mutations on pH gating. The ordinate in Fig. 3h does no report Fractional inhibition, but rather Fraction of current remaining (upon acidification). However it would suffice to describe the data as $I(\text{pH } 5.5)/I(\text{pH } 7.5)$.

Otherwise the results are clearly presented.

Response to reviewer comments

We would like to thank the reviewers for their time, careful reading, and constructive feedback on our manuscript. We have addressed all comments in the revised manuscript and point-by-point response below. We hope you will agree the paper is substantially improved as a result.

Reviewer #1 (Remarks to the Author):

This manuscript describes structural studies of the K2P channel K2P1.1 (TWIK1) using cryo-electronmicroscopy. The authors determine structures of TWIK1 in high (pH 8.0) and low (pH 5.5) conditions that should correspond to the active and inactive forms (when the channel is under high potassium concentrations). The structures reveal that the low pH conditions induce a conformational change in the selectivity filter that involves flipping out of key residues, zippering of the filter, loss of ions in the outer sites, constriction of the extracellular ion pathway (EIP) by the upward movement of the cap, and possible block of the ion pathway by lipids that enter the central cavity through lateral caps under the selectivity filter. Together, these changes appear to show another twist on the ways that selectivity filter conformational changes can impact potassium channel function. The general area of 'C-type' gating is now receiving a great deal of attention from a variety of structural studies and as such the results here are timely and important as the further show the remarkable structural plasticity of potassium channel selectivity filters. Nevertheless, the manuscript has a number of key deficits in terms of data quality, presentation, and contextualization and reference to prior studies that need to be addressed.

Major issues:

Two electrode voltage clamp recordings show a surprising variability regarding the reversal potential. These suggest suggest either poor quality oocytes or important selectivity changes. Reversal potentials for WT and the mutations shown in Fig. S5 are varied, ranging from less than zero (WT), zero (T123G, T123D), to substantially above zero (D230N). These changes suggest either that there are substantial alterations to ion selectivity, or that there was poor control of the background leak from the oocytes. The authors even note that the D230N has no effect on pH, implying that this is a wild-type channel. Clearly, if the reversal potential in S5A is correct, D230N is not a wild-type-like channel. Further, in terms of comparison with other published studies in oocytes using similar conditions (ex. Chatelain et al. PNAS 109:5499 (2012), the WT value appears to be more negative than expected. These issues need to be addressed, especially as there was no comment on these apparent fundamental changes in the properties of the studied channels.

We have clarified the text and included new data in Fig. S5 to address this point. Reversal potentials for each mutant in low $[K^+]_{\text{ext}}$ (96 mM Na^+ , 2 mM K^+ , pH 8) and high $[K^+]_{\text{ext}}$ (2 mM Na^+ , 96 mM K^+ , pH 8) are now shown and poor quality recordings have been removed. Some mutants indeed show altered K^+ vs. Na^+ selectivity with different reversal potentials in low $[K^+]_{\text{ext}}$ (reversal potentials in high $[K^+]_{\text{ext}}$ are not statistically different, presumably because expected differences are small under these conditions). This is consistent with previous work showing that mutations in these amino acid positions (or interacting positions) alter selectivity in other channels (Sauer et al. PNAS 2011; Cheng et al. PNAS 2011). We state that our conclusions are based on comparing pH sensitivity of mutants that display altered selectivity in some cases.

Reversal potentials for wild-type TWIK1 (in 2 mM or 96 mM $[K^+]_{\text{ext}}$) are similar to those previously reported (Chatelain et al. PNAS 2012, Fig. 4B). Both are near E_{K^+} implying high K^+ selectivity, as expected.

Although the data appear to provide a clear model for how the channel is inhibited by low pH under high potassium conditions, the authors make no attempt to reconcile how this sort of state, which seems to be inhibited three ways 1) closing of the filter, 2) narrowing of the EIP by movement of the cap, and 3) the proposed block by a cavity lipid, could be relevant for the dramatic the loss of selectivity seen at low pH / low $[K^+]$. In general, the authors do not do a good job at putting the results here in the context of other studies of K2P filter gating, in terms of how this low pH/high $[K^+]$ inhibited state is or is not related to the ideas of selectivity filter instability (Nematian-Ardestani et al. J Biol. Chem. 295:610-618 (2020)), or why the low pH / low $[K^+]$ state is not only active but allows the passage of sodium ions. As the authors show that their construct undergoes this selectivity loss (Fig. 1B), there should be some sort of discussion on this

point. Given the seemingly tightly zippered selectivity filter with additional cavity lipids occluding the filter from below, it does not look like Na⁺ ions would be able to pass through this structure without additional conformational changes - especially in regard to the 'lipid block' which looks like a total occlusion. It seems likely that the leaky state at low [K⁺] and low pH is something quite different than what is observed here.

We agree, for all these reasons, that TWIK1 must adopt a different structure to account for conductance and selectivity loss under low pH/low [K⁺]_{ext} conditions. We do not know what conformational changes are involved. We have added a section to the discussion to make this explicit. We also reference the idea of filter instability (Nematian-Ardestani et al. 2020), reference the unusual threonine in the TWIK1 selectivity filter critical for selectivity changes, and note that a cavity created by movement of H122 and T123 behind the selectivity filter in the closed structure may contribute to filter instability.

In the 2nd to last paragraph on page 4 of the pdf which starts "Selectivity filter gating..." they compare results from other studies in regards to how the next residue just outside the filter flips up. The authors write: "In TREK1, N147 flips up in response to changes in [K⁺]_{ext}, although this conformational change is only observed within one subunit and does not fully block the conduction path (Lolicato et al, 2020)." This is a misleading statement. While it is true the N147 flip is only in one subunit in the TREK1 crystal structures, the simulations Lolicato et al., 2020 indicate that this conformational change happens in both subunits (as shown in the symmetrized models in Fig. 6 of Lolicato 2020) and blocks the conduction path. Of note, such interpretations in which changes in one subunit are placed in the context of a symmetric homodimer are in line with prior K2P studies in which movement is observed by crystallography in only one subunit, but symmetrized versions were displayed and used to develop mechanistic models (ex. Brohawn et al. Nature 2014). The authors should note both the structural and simulation studies that support a symmetrical change at the N147 site in K2P2.1 (TREK-1).

Agreed and thank you for pointing out the misleading wording. We have made these corrections.

Both the presentation and discussion of the potential blocking lipid is problematic. The evidence for the lipid is not presented when the concept is first introduced. The block is noted in the cartoon in Fig. 2b, but the data that support this model are in Fig. S7g, which are only introduced much later in the manuscript. Further, from the view in Fig. S7g, it is unclear if the lipid density merges with the density for the ion or is at a different altitude along the channel central axis. The authors also present the lipid block as being equivalent to norfluoxetine. Although the site may be the same, norfluoxetine does not block the inner cavity. This is an important distinction that needs to be noted as the authors seem to imply a common mechanism, where one does not exist.

We addressed these issues with the following three changes.

The pore analysis in Figure 2 is intended to provide the reader with an overall impression of the differences between structures and to indicate the three positions where conduction is obstructed at low pH prior to explaining the structural basis of each obstruction in detail. We prefer this way of presenting the data over showing detail for each before the overall view. We now state that data supporting each constriction is discussed in turn.

We have improved Figure S7 to make clear that the lipid binding site is at a lower altitude than the selectivity filter ion.

We have clarified our intended point that the cavity and lateral membrane opening/fenestration have been implicated in modulation of several K2Ps with lipid block and drug modulation accomplished through distinct mechanisms.

The question of lipid access to the K2P central cavity remains a controversial one. Tucker et al. have ruled out such mechanisms from both data and simulations (McClenaghan et al. J. Gen. Physiol. 147:497-5050 (2016), Aryal et al. Structure 25:708-718 (2017)). One cannot exclude from the current data that what the authors see here is the access of residual detergent that enters the cavity in the way that a lipid that is part of the membrane bilayer cannot. The authors are certainly free to offer their own view of their data, but they should address the work of Tucker et al. in the discussion of possible lipid block mechanisms and potential shortcomings. Further, as noted above, the low pH lipid block would seem to be a major impediment to the sodium flux that happens in the low pH/low [K⁺] state and providing some comment on this seems important.

We now include a section stating that the physiological relevance of lipid block of K2Ps is debated and understanding the role of lipid block in TWIK1 requires further investigation. We cite work of Tucker et al. related to TREK2 gating (as suggested), work of Tucker et al. related to TWIK1 gating, and our work related to TRAAK.

We now state we cannot exclude the possibility that the acyl chain corresponds to a residual detergent molecule. We also list reasons why we think this is unlikely. "(1) purification and nanodisc reconstitution took place under conditions of high pH that favor low acyl chain occupancy prior to a final chromatography step at low pH without detergent, (2) all three lipid species used in the nanodisc reconstitution could reside in a low energy conformation with a headgroup in the inner leaflet and an acyl chain bound in the channel cavity as observed based on geometric and steric considerations, and (3) lipid acyl chains are present at higher concentrations in the nanodisc relative to potential residual detergent and would likely have higher fractional occupancy in the loose hydrophobic binding site of the channel cavity."

As noted above, we agree Na⁺ permeable TWIK1 conformations must correspond to other unknown structures and state this in the discussion.

Statement that the work here first example of the cap being involved in K2P gating is not true. The cap has been shown to be intimately involved in channel gating by poly ruthenium amines and other cations (Gonzalez et al. J. Biol. Chem. 228 5984-5991 (2013), Braun et al. Br. J. Pharmacol. 172:1728-1738 (2015), Pope et al. Cell Chem Biol 27:511-524.e4 (2020)). The movements here have essentially similar structural consequences to modulation by these polycations, namely, physical block of ion access to the EIP. While these structures may be the first example of cap movement being linked to gating, the authors should rephrase their description to acknowledge the prior roles of the cap shown by others and note that there are conceptual links to what they observe here. Accurately expanding this description will improve the generality of their observations and take nothing away from the novel conformational changes they observe here.

Thank you for pointing out this ambiguity in our presentation. Our intent was to point out this is the first example of helical cap movement involved in gating. These citations have been added as examples of how helical caps have been implicated in modulation without large scale movement.

Minor points:

The authors should (at least in the abstract and introduction) use the regularized name for this channel, K2P1.1 (TWIK-1) (and any other K2P channels that are discussed). The authors should include these names (recognized by the International Union of Basic and Clinical Pharmacology. <http://journals.ed.ac.uk/gtopdb-cite/article/view/6417>) to ensure that readers are clear about which channels are under discussion here. It is also important for various literature searches that this official name is found in the manuscript, even if the authors then choose to revert to 'TWIK1' for the rest of their writing.

"K2P1.1" has been added to the abstract and introduction of the manuscript and regularized/IUPHAR recognized names are included with the first instance of the descriptive name of all K2Ps.

In Figs. 3 f & g, the structure is labeled SF1, but based on the amino acids present it appears to be SF2.

Corrected. Thank you for noting this error.

When the authors compare the changes here with other C-type gating changes, it would seem appropriate to cite the work for the various examples upon first mention (i.e. in the line 'reminiscent of upward movements by analogous residues in the K2P TREK1 (ref), the archaeal channel KcsA (ref), (Note: KcsA does not come from an archaeon but from a gram-positive bacterium. Please correct this)...the voltage-gated K⁺ channel Shaker (ref).

Corrected.

There seems to be an error in the third paragraph of the introduction 'add Aryal et al'. It looks like this was a 'note to self' during the drafting of this manuscript and represents an incomplete edit.

Corrected.

References to the relevant papers should be made along with the PDB codes in Figure S9.

These references have been added.

The Supplementary movies were not part of the review package. I assume that they are identical to those posted on bioRxiv (<https://www.biorxiv.org/content/10.1101/2021.11.09.467928v1.supplementary-material>) If so, the movies are fine and clearly present the motions under discussion.

Indeed, these are correct and were meant to be included. Thank you.

Reviewer #2 (Remarks to the Author):

TWIK1 is a K2P channels that is highly expressed in the brain and heart and plays an important role in cardiac rhythm generation and insulin release. It is gated by pH, closing at pH 5.5 and opening at pH 7.5. Although a x-ray structure of TWIK1 at high pH has been published, the pH-dependent gating of TWIK1 remains unknown. In this manuscript, Turney et al. determined cryo-EM structures of TWIK1 at high and low pH in nanodisc at resolutions up to 3.4Å. They observed different conformations at different pH – at high pH, TWIK1 is in an open conformation while at low pH it changes to a closed conformation. Comparing these two structures revealed a cascade of conformational changes throughout the protein, providing a gating mechanism of TWIK1 induced by pH. They observed a conformational change in the selectivity filter from high pH to low pH and elucidated an interesting gating mechanism associated with selectivity filter. Since TWIK1 has unique properties in K2Ps, the results presented by the authors provide a mechanistic understanding of its unique function properties. Overall, this is an interesting and important study of K2P channels that will deepen our understanding on the pH-dependent gating mechanism of TWIK1. While the structural and functional studies here are solid, I do have several comments that the authors need to address before the manuscript can be accepted.

1. The manuscript has some readability issues. First, the figures are not properly labeled. For instance, the “helical cap” are mentioned throughout the text, but are not labeled in the figures. The same is true for the “SF2-TM4 linker” or “SF1-TM2 linker” etc. The authors should read through the text and figures to make sure that everything is labeled properly. Second, the Figures are not properly referenced throughout the text. For instance, there are eight panels in Figure 3, but the authors only cited “Figure 3” in the text without citing each panel. This makes it difficult for the readers to connect the text to the figures; Another example is the paragraph on page 2 “At high pH, the selectivity filter adopts a nearly four-fold symmetric ...”, where no figure was cited even though extensive conformational changes were described. Third, a considerable amount of conformational changes described in the text are not shown (or can't be found) in the figures. For instance, “H122 flips upward towards the helical cap. The adjoining linker region connecting SF1 to TM2...”. Overall, the authors should have done a better job of improving readability.

We thank the reviewer for pointing out the difficulty connecting text and figures. We have added labels and figure references throughout the manuscript as suggested.

2. It is unclear how can the clash happen in “In its pH 5.5 position, H122-L126 clashes with E84-G89 at the bottom of EC2”, and how it is linked to the gating mechanism. Please clarify.

We have clarified the text to indicate H122-L126 in the low pH position would clash with E84-G89 in the high pH position, so E84-G89 must undergo conformational changes (relative to its low pH position) to prevent steric overlap.

3. What is the gray molecule in Figure S7c?

The cavity-bound acyl chain (now labeled). Thank you for pointing out this omission.

4. Why the crystal structure of TWIK1 at high pH adopted the same conformation at low pH?

We cannot say definitively but offer several possibilities (now made more clear in the text): it represents an infrequently populated TWIK1 conformation at high pH favored by crystal contacts, differences in protein construct or solution conditions promote different conformations, and/or differences in solubilizing

environment (detergent micelle vs. lipid nanodisc) promote different conformations.

Reviewer #3 (Remarks to the Author):

K2P potassium channels provide the background conductance of all cells. They are exquisitely regulated by a variety of stimuli and mutations affecting them are the cause of several human diseases. TWIK-1 is the founder member of the K2P family and has been extensively studied. TWIK-1 is gated by pH through a known pH sensor, an extracellular facing histidine residue, but details of the molecular mechanism involved are not known. More generally, the molecular mechanisms of gating of the K2P channels is only beginning to be understood.

This paper addresses the mechanism of gating of TWIK-1 K2P potassium channel by pH through the resolution by cryo electron microscopy of the structures of TWIK-1 in a lipid environment at neutral and acidic pH, which correspond to conditions under which the channel is open and closed respectively.

The structures reveal a complex mechanism of gating. The channel exhibits a continuous open permeation pathway at neutral pH while at acid pH a change in orientation of the sensing histidine is accompanied by alterations at the extracellular access to the pore and the upper reaches of the selectivity filter prohibiting potassium flow. Remarkably, the closed state is also associated with intramembrane fenestration changes allowing lipid entry and further inner pore blockade of the permeation pathway.

The results give important novel ideas of the mechanisms of gating in K2P channels. They offer the first molecular description of widespread C-type inactivation in K2P channels. It also reaffirms the importance of lipid intrusion in gating and provides a function for intramembrane fenestrations in K2P channels as gated structures.

The work is original and clearly of high significance for our understanding of K2P channel regulation. The methodology employed is sound and the authors' laboratory are leaders in the field. The presented evidence fully supports the contentions of the paper and there no caveats in the data analysis. The paper is succinct but data are well presented.

A minor point concerns a problem with the presentation of functional results of the effect of mutations on pH gating. The ordinate in Fig. 3h does no report Fractional inhibition, but rather Fraction of current remaining (upon acidification). However it would suffice to describe the data as $I(\text{pH } 5.5)/I(\text{pH } 7.5)$.

Thank you for pointing out this error. The axes in Figs. 1A and 3H have been corrected to $I(\text{pH } 5.5)/I(\text{pH } 8.0)$.

Otherwise the results are clearly presented.

REVIEWERS' COMMENTS

Reviewer #1 (Remarks to the Author):

The authors have done a good job of addressing the critiques. Upon re-examining the manuscript, I identified a few other issues that still should be addressed prior to publication.

While going through the PDB reports, I realized that a good fraction (at least half of the constructs) are missing density. These missing elements appear to be largely on the N- and C-terminal ends. It would be helpful to the reader if the authors include a figure showing the sequence on the construct with the resolved/ unresolved elements indicated (ex. Fig. S2 of Brohawn et al., Science, 2012, but only showing the relevant sequence here for the structures).

A related complication is that the two structures have differences in which parts of the protein are resolved. This is most apparent in Fig. 4 in which one can see that the N-terminal end of the pH 7.4 structure is missing ~ one helical turn (The pH 7.4 starts at residue 20, whereas the pH 5.5 structure starts at residue 24). The ending residues should be labeled in Fig. 4. Although this information is stated in the text, without providing a clear graphical guide to what is resolved, what is not resolved, and what is different between the two structures, I am concerned that readers, especially those not ultra-savvy with structures, will miss these key points or get confused, or might think that there is some sort of conformational difference (which formally there is, as this element clearly has different properties in the two structures) in this region.

The authors should update the reference to the Tan 2021 study as this is now published in Science Advances.

Reviewer #2 (Remarks to the Author):

The authors have addressed all my concerns, I am satisfied with their responses.

Reviewer #3 (Remarks to the Author):

The authors have thoroughly and ably responded to all concerns I and the other reviewers had raised. Extensive amendments to the manuscript have raised the quality of the paper significantly.

The findings reported advance our knowledge of gating mechanisms of K2P channels and provide a basis for further work to understand the function of this important class of ion channels in a physiological setting.

Response to reviewer comments

We have updated the manuscript to address the small issues raised by reviewer 1 as indicated below.

Reviewer #1 (Remarks to the Author):

The authors have done a good job of addressing the critiques. Upon re-examining the manuscript, I identified a few other issues that still should be addressed prior to publication.

While going through the PDB reports, I realized that a good fraction (at least half of the constructs) are missing density. These missing elements appear to be largely on the N- and C-terminal ends. It would be helpful to the reader if the authors include a figure showing the sequence on the construct with the resolved/ unresolved elements indicated (ex. Fig. S2 of Brohawn et al., Science, 2012, but only showing the relevant sequence here for the structures).

A related complication is that the two structures have differences in which parts of the protein are resolved. This is most apparent in Fig. 4 in which one can see that the N-terminal end of the pH 7.4 structure is missing ~ one helical turn (The pH 7.4 starts at residue 20, whereas the pH 5.5 structure starts at residue 24). The ending residues should be labeled in Fig. 4. Although this information is stated in the text, without providing a clear graphical guide to what is resolved, what is not resolved, and what is different between the two structures, I am concerned that readers, especially those not ultra-savvy with structures, will miss these key points or get confused, or might think that there is some sort of conformational difference (which formally there is, as this element clearly has different properties in the two structures) in this region.

We have updated Supplementary Figure 10 to address these issues. The requested graphical key is now included. C-terminal residues are indicated in Fig. 4.

The authors should update the reference to the Tan 2021 study as this is now published in Science Advances.

Updated. Thank you.